# Use of the Thyromental Height Test for Prediction of Difficult Laryngoscopy: A Systematic Review and Meta-Analysis

**DOI:** 10.3390/jcm11164906

**Published:** 2022-08-21

**Authors:** Wenxuan Chen, Tian Tian, Xintao Li, Tianyu Jiang, Fushan Xue

**Affiliations:** 1Sixth Clinical Medical College and Beijing Anzhen Hospital, Capital Medical University, Beijing 100054, China; 2Department of Anesthesiology, Beijing Friendship Hospital, Capital Medical University, Beijing 100050, China

**Keywords:** thyromental height test, laryngoscopy, airway management, systematic review, meta-analysis

## Abstract

The thyromental height test (TMHT) has been proposed as a novel single clinical test for predicting difficult laryngoscopy (DL), though consequent studies have put forward various estimates when verifying its reliability. This systematic review and meta-analysis aimed to provide a comprehensive evaluation of the predictive value of TMHT for DL. A computerized search of CNKI, CQVIP, EBSCO, PubMed, SinoMed, and Wanfang Data was conducted on 1 June 2022. Prospective cohort studies reporting diagnostic properties of TMHT in relation to Cormack and Lehane grading in patients aged more than 16 years, either sex, scheduled for surgery under general anesthesia, requiring tracheal intubation with direct laryngoscopy were included in this analysis. Data was extracted or calculated, and meta-analysis was done by the Stata MIDAS module. A total of 23 studies with 5896 patients were included in this analysis. Summary estimates of all included studies are as follows: sensitivity 74% (95% CI, 68–79%); specificity 88% (95% CI, 81–92%); diagnostic odd ratio, 20 (95% CI, 10–40); positive likelihood ratio, 5.9 (95% CI, 3.6–9.6); and negative likelihood ratio, 0.30 (95% CI, 0.23–0.39). Summary sensitivity and specificity for studies with a prespecified threshold were 82% (95% CI, 71–89%) and 94% (95% CI, 87–98%), respectively. The estimated area under curve (AUC) was 85% (95% CI, 81–88%). There was no significant threshold effect but significant heterogeneity in both sensitivity and specificity. Heterogeneity in sensitivity became insignificant after removing two outliers of sensitivity analysis. It is concluded that THMT has an overall optimal predictive value for DL in adult patients with diverse ethnicity and various risk factors, displaying better predictive values in a large patient population comparing to other recent reported bedside assessments and a previous meta-analysis. As significant heterogeneity brought by un-standardized application of external laryngeal manipulations in the included studies may have biased the results of this meta-analysis, the actual predictive value of TMHT for DL still awaits further studies with good designs and large sample sizes for better determination.

## 1. Introduction

A difficult airway is the clinical situation in which a conventionally trained anesthesiologist experiences difficulty with facemask ventilation, laryngoscopy and intubation, supraglottic airway ventilation, extubation, or invasive airway [1]. Adverse airway events resulting from unanticipated difficult laryngoscopy (DL) or intubation (DI), such as airway injury, esophageal intubation, and aspiration, are major causes of anesthesia-related perioperative morbidity and mortality [2,3]. Although not exactly equivalent to DI, DL is currently the reliable clinical predictor that signals a warning for high risk of DI. Thus, a reliable airway assessment test with a high accuracy as the predictor for DL is vital for the safety of clinical anesthesia.

Most airway assessment tests, such as upper lip bite test (ULBT), modified Mallampati test (MMT), and thyromental distance (TMD), have been recommended for preoperative prediction of DL, but recent robust evidence indicates that no any single airway assessment test can reliably predict the occurrence of DL [4,5,6,7]. To improve predictive accuracy, different combinations of airway assessment tests have been suggested [4,8,9]. A recent study showed that the combination of ULBT and MMT had the best predictive ability for DL, with a sensitivity of 88.9% and a specificity of 93.2% [10]. Thus, it is still necessary to continuously explore airway assessment tests with a good predictive ability for DL.

The thyromental height test (TMHT) was first proposed as a clinical test by Etezadi et al. in 2013 [11], and it showed a surprisingly high predictive value for DL at a 5 cm threshold based on a relatively small sample study. In the original study of Etezadi et al. [11], thyromental height (TMH) was defined as the height between the anterior border of the thyroid cartilage (on the thyroid notch just between the 2 thyroid laminae) and the anterior border of the mentum (on the mental protuberance of the mandible), with the patient lying supine with her/his mouth closed. Subsequently, several studies evaluated the actual performance of TMHT as a single predictor for DL in different populations [11,12,13,14,15,16,17,18,19,20]. In 2021, moreover, Carvalho et al. [21] performed a meta-analysis that included eight studies and showed that TMHT was a good predictor of DL with a better performance than most previously reported bedside airway assessment tests. In the meta-analysis of Carvalho et al. [21], however, exclusion of non-English language studies may have resulted in the absence of some important studies. Most importantly, several new works assessing the performance of TMHT for prediction of DL have been published after their meta-analysis [22,23,24,25,26,27]. To further determine the actual performance of TMHT as a single predictor of DL, we performed this systematic review and meta-analysis including all 23 studies in the available literature.

## 2. Methods

### 2.1. Protocol and Registration

This systematic review and meta-analysis of diagnostic test accuracy was designed, conducted, and reported in accordance with the Preferred Reporting Items for Systematic reviews and Meta-Analyses (PRISMA) guidelines [28,29]. The review protocols had been designed before literature screening, registered at PROSPERO (http://www.crd.york.ac.uk/PROSPERO, accessed on 1 April 2022, registration number: CRD42022319323), and followed throughout the entire work.

### 2.2. Eligibility Criteria

For this systematic review of diagnostic test accuracy, we included only studies of prospective cohort design or randomized controlled trials with full reports which met the following criteria: (1) Languages: studies published in languages restricted to English, Chinese, and Portuguese; (2) Populations: patients recruited were ≥16 years of age, either sex, scheduled for surgery requiring endotracheal intubation by direct laryngoscopy under general anesthesia. No restrictions were placed on patients’ American Society of Anesthesiologists (ASA) physical status classification, other specific health conditions, the healthcare setting, or the healthcare professionals involved.

Index test: a study was included as long as both the measuring method and reported data for TMHT were consistent and complete for all patients. However, the studies about the predictive accuracy of modified thyromental height test (MTMHT) for DL were excluded, as TMH was measured in a similar but different manner.

Reference standard test: The laryngoscopy view of glottis was determined by the classical Cormack and Lehane (CL) grading system, where grade 3 (only epiglottis visible) and grade 4 (neither glottis nor epiglottis visible) are considered as DL [1]. Studies presenting data on DL based on other tests and other ranges of CL grading systems were excluded.

Data: Numbers of cases for true positive (TP), false positive (FP), false negative (FN), and true negative (TN) were reported, respectively; otherwise, sensitivity and specificity along with total sample size and the number of DL should be provided for manual calculation.

### 2.3. Information Sources and Search Strategy

The following databases were searched on 1 June 2022: CNKI, CQVIP, EBSCO, PubMed, SinoMed, and Wanfang Data. Reference lists of included studies were also searched and those potentially relevant to TMH were retrieved.

### 2.4. Selection Process

First round screening of literature was performed merely on titles and abstracts for relevancy, followed by a second round screening, where full texts of all remaining papers were assessed against eligibility criteria and study quality. Both rounds of screening were conducted independently by two reviewers (WXC and TT). Uncertainties and disagreements were resolved by their discussion.

### 2.5. Data Collection Process and Data Items

One reviewer (WXC) independently extracted and calculated the data of interest through a standardized form in Microsoft Excel from each included study, and it was then verified by another reviewer (TT). Uncertainties and disagreements were resolved by their discussion. The key items included in the data chart are: authors, year of publication, design of study, age, gender, height, weight, body mass index (BMI), total sample size, sample size of DL groups, mean TMH, and cut-off values of TMH, TP, FP, FN, TN, sensitivity, and specificity. When data for more than one threshold or laryngoscopy manipulations were provided in a single study, they were considered as different data groups and displayed separately.

### 2.6. Study Risk of Bias Assessment

Both risk of bias and applicability concerns were assessed independently by two reviewers (WXC and TT), using a revised tool for the Quality Assessment of Diagnostic Accuracy Studies (QUADAS-2) [30]. Each study was coded as ‘high’, ‘low’, or ‘unclear’ risk/concern, according to the corresponding answers of several signaling questions about: (1) patient selection; (2) index test; (3) reference standard, and (4) patient flow and timing. The answers could be chosen from ‘yes’, ‘no’, and ‘unclear’, where a single ‘no’ leads to ‘high’ risk/concern, and only ‘yes’ for all questions leads to ‘low’ risk/concern. The assessing process was conducted in the Review Manager (RevMan, London, UK, v5.3.5) [31]. Uncertainties and disagreements were resolved by discussion.

### 2.7. Diagnostic Accuracy Measures

Sensitivity and specificity of TMHT for DL, which was defined by grades 3 and 4 of the CL grading system, were the primary outcomes of this study. Diagnostic odd ratio (DOR), positive likelihood ratio (LR+), and negative likelihood ratio (LR−) were also calculated for further detailed analysis.

### 2.8. Synthesis Methods

The statistical analysis for this study was performed in Stata (StataMP, release 16, Lakeway, TX, USA) with the module for meta-analytical integration of diagnostic test accuracy studies (MIDAS) [32]. Diagnostic properties, including TP, FP, FN, and TN were either collected or calculated, which enabled the production of forest plots for sensitivity and specificity of TMHT for diagnosis of DL. Forest plots for diagnostic odd ratio (DOR), positive likelihood ratio (LR+), and negative likelihood ratio (LR−) were also depicted. Overall heterogeneity was evaluated by the Cochran’s Q test along with the Spearman correlation test for the presence of diagnostic threshold effect. When heterogeneity was present (*I*^2^ > 50%), sensitivity analysis was performed and a forest plot for estimates was built to evaluate the contribution of each study to the overall heterogeneity. Studies that were the most responsible for heterogeneity were then eliminated before further analysis. Summary receiver operating characteristic curves (SROC) were also generated in Stata MIDAS module [32] by which summary sensitivity and specificity, and the area under curve (AUC) were calculated. Furthermore, summary sensitivity and specificity were estimated for studies with a same TMHT cut-off value (5 cm) and with a prespecified cut-off value.

### 2.9. Publication Bias Assessment

Publication bias was assessed by the Deek’s funnel plot asymmetry test in Stata with MIDAS module, which performed the linear regression of log odds ratios on inverse root of effective sample sizes as a test for funnel plot asymmetry. A *p* value of less than 0.10 was set for the significance threshold [32].

## 3. Results

### 3.1. Study Selection

After computerized research through several databases, 93 papers were identified, but only 54 remained after eliminating 39 duplications. A PubMed search strategy is displayed in Appendix A. First round literature screening was conducted on titles and abstracts of these 54 articles, 26 of which were further excluded for to two reasons: (1) retrospective evaluation, literature review, and letter (6 articles); and (2) irrelevance to our study objectives (20 articles). Full texts of all 28 remaining articles were retrieved before the second-round screening process. Careful assessment was performed with thorough reading and application of the eligible criteria. As a result, a total of 5 articles were excluded because of the following reasons: (1) DL identification methods other than the CL grading system (2 articles); (2) missing or inconsistent data (2 articles); (3) different measuring procedures for TMH (1 article). A PRISMA diagram [28] for the complete study selection process is shown in Figure 1. After the study selection process, all included studies were in either English or Chinese.

### 3.2. Study Characteristics

Important characteristics of each included study are summarized in Table 1. Among 23 independent studies including 5896 patients, a total of 615 patients were reported as DL according to the CL grading system. The incidence of DL in the included studies ranged from 1% to 31%. These prospective cohort observational studies took place in Australia, Bangladesh, China, Egypt, India, Iran, Japan, Nepal, and Turkey. Patients undergoing elective surgeries under general anesthesia requiring tracheal intubation were recruited. All studies included only patients without obvious airway abnormalities and malformations. One study [12] included only patients scheduled for coronary artery bypass surgery, two [19,26] only elderly patients (≥65 years), and another two [22,25] only obese patients with a BMI > 30 kg/m^2^.

As for the TMH measurement during preoperative assessment, 17 studies mentioned the use of either digital gauges (10 articles) [11,12,13,16,18,20,22,25,26,27] or regular rulers (7 articles) [14,15,17,19,23,24,33], while 6 studies failed to specify their measurement tools. Patients were placed in sniffing position for direct laryngoscopy and intubation in 14 studies [11,12,14,16,17,18,19,20,23,26,27,33,34,35]. During the direct laryngoscopy, a Macintosh blade was used in 17 studies in which one [31] used only size 3, 7 [12,16,17,18,20,23,26] used size 3 or 4, 2 [11,25] used only size 4, and 1 [13] used size 4 or 5. Only one study [11] mentioned the use of a Miller blade instead of a Macintosh when no laryngeal view was achieved and a second attempt was needed. Almost all laryngoscopy procedures were conducted by experienced anesthesiologists, except for one study [11] by residents and two [36,37] without mentioning. Application of external laryngeal manipulation showed inconsistency among studies, and its application or applicable condition lacked clear statements in most of the studies. The CL grading system was applied in all studies as for the eligible criteria. The CL grade 3 or 4 was the most approved diagnostic standard for DL, while only two [16,26] used the CL classification 2b or higher as their standard for DL.

### 3.3. Risk of Bias in Studies

As displayed in Figure 2 for methodological quality assessment by the QUADAS-2 tool, risk of bias in individual studies mainly came from patient selection, index test, and reference standard. Inappropriate exclusion criteria proposed in 13 studies [12,14,15,16,18,20,22,23,24,25,27,34,37] accounted for the high risk of bias in terms of patient selection. There was high concern that the test accuracy reported by these 13 studies could be positively affected by the fact that they removed patients with obesity, pregnancy, and other factors potentially increasing the possibility of DL. The provenance of bias related to the index test was straightforward: inability to preset a diagnostic TMHT threshold [11,12,13,17,18,19,22,25,26,27,33,36,37]. Bias regarding the reference standard was basically due to the absence of blindness [12,14,22,24,25,36,38,39] and the un-standardized application of external laryngeal manipulation [11,12,16,18,19,20,22,26,36]. All studies showed a low risk/concern for flow and timing and applicability.

### 3.4. Results of Syntheses

The Stata codes used for this meta-analysis are displayed in Appendix A. All figures were direct outputs of Stata (StataMP, release 16) [32] and Review Manager (RevMan, London, UK, v5.3.5) [31]. As shown in Figure 3, sensitivity and specificity of TMHT for prediction of DL reported in all 23 studies ranged from 39% to 95%, and 53% to 100%, respectively. The ranges for other diagnostic accuracy measurements were as follows: DOR, 1.33 to 721.29 (Figure 4A); LR+, 1.17 to 149; LR−, 0.05 to 0.87 (Figure 4B).

Analysis was first conducted with data from all included studies, resulting in a summary sensitivity of 74% (95% CI, 68–79%) and a specificity of 88% (95% CI, 81–92%) (Figure 3). Other summary estimates included: DOR, 20 (95% CI, 10–40) (Figure 4A); LR+, 5.9 (95% CI, 3.6–9.6); and LR−, 0.30 (95% CI, 0.23–0.39) (Figure 4B). The estimated area under curve (AUC) for the SROC curve was 85% (95% CI, 81–88%) (Figure 5).

After removing two studies [18,27] of high heterogeneity, the same analytical procedure was conducted again, which will be elaborated in the next section. The summary sensitivity and specificity were 77% (95% CI, 72–81%) and 90% (95% CI, 84–94%), respectively, after their removal (Appendix A).

A total of 14 studies [11,12,13,14,15,16,17,18,20,23,24,27,38,39] with the same TMHT threshold (5 cm) showed a summary sensitivity of 75% (95% CI, 66–83%) and a specificity of 91% (95% CI, 82–95%) (Appendix A). A total of 10 studies [14,15,16,20,23,24,34,35,38,39] with prespecified TMHT thresholds showed a summary sensitivity of 82% (95% CI, 71–89%), a specificity of 94% (95% CI, 87–98%) (Appendix A), and an AUC of 92% (95% CI, 90–94%) (Appendix A).

### 3.5. Reporting Biases

*Threshold effect:* There was no a significant threshold effect building up to the heterogeneity of this study (Spearman correlation estimate 0.70, *p* = 0.49).

*Heterogeneity and**sensitivity analysis:* There were significant heterogeneities in both sensitivity (*p* < 0.001, *I*^2^ = 78.42) and specificity (*p* < 0.001, *I*^2^ = 98.60) of TMHT for prediction of DL (Figure 3). Sensitivity analysis was therefore conducted, and it showed two main sources of heterogeneity [18,27] (Appendix A). After removing the data of these two studies and repeating the Cochran’s Q test, the heterogeneity in sensitivity (*p* = 0.001, *I*^2^ = 44.48) was no longer significant but the specificity (*p* < 0.001, *I*^2^ = 98.51) remained the same. Other possible factors contributing to the heterogeneous significance might be related to the concerns for the reference standard as previously mentioned in methodological quality assessment. Different TMHT thresholds, preoperative threshold specification, standardization of external laryngeal manipulation (backward, upward, rightward pressure, BURP), blade sizes, neuromuscular blockage, and operators’ experience were the potential candidates for our heterogeneity.

*Publication bias:* Deek’s funnel plot asymmetry test provided a chance for visual inspection and statistical calculation at the same time, both suggesting a low risk of publication bias (*p* = 0.19) (Appendix A).

## 4. Discussion

The prediction of difficult airways has always been a crucial task for anesthesiologists in terms of airway management. Systematic reviews have been performed on various preoperative assessment methods, but inconsistent conclusions have been drawn [40,41,42]. According to the summary estimates of our review and analysis, the overall predictive value of TMHT for DL seems optimistic, with a sensitivity of 74%, a specificity of 88%, and an AUC of 85%. For all that, uncertainties are still present and should be addressed with caution. Especially, significant heterogeneity, relatively large 95% CI, and un-standardized application of external laryngeal manipulation during laryngoscopy are the potential weaknesses that need further discussion.

Among all 23 studies, the reported incidence of DL varied from 1% to 31%. Furthermore, the reported sensitivity and specificity of TMHT for prediction of DL ranged from 39% to 95%, and 53% to 100%, respectively. Although significant variability in sensitivity and specificity was reported, TMHT had an overall impressive specificity, and high sensitivity in these included studies. In total, 10 out of 23 studies had a specificity of above 90% [11,12,15,16,17,22,23,24,34,35], emphasizing an outstanding value of TMHT in differentiating non-DL patients from the others. In total, 16 [11,12,13,15,16,17,19,20,22,23,24,25,26,33,34,35] out of 23 studies reported a sensitivity of more than 70% comparing to the CL grading system, and 3 [13,15,24] among them over 90%, depicting a rather promising prediction of true DL. When a 5 cm threshold was set in the study, as proposed by Etezadi et al. [11], increased sensitivity and specificity were obtained, indicating the rationality and necessity of a 5 cm threshold.

This analysis showed that compared to other major predictors studied in recent literatures [5,7,40,41,42], TMHT had a satisfying predictive potential for DL with stability, comprehensiveness and independence. A systematic review and meta-analysis on various airway ultrasound predictors, such as the distance from skin to epiglottis (DSE), the distance from skin to hyoid bone (DSHB), and the distance from skin to vocal cords (DSVC), showed that DSE was the best imaging predictor, with a sensitivity of 82% (95% CI, 74–87%), a specificity of 79% (95% CI, 70–87%), and an AUC of 87% (95% CI, 84–90%) [40]. As patients with a history of previous difficult intubation or expected difficult laryngoscopy have been excluded from the above analysis of airway ultrasound predictors, the overall quality of evidence is low/very low and there is a high concern of bias [40]. In our analysis, however, TMHT demonstrated a higher specificity, which is the ability to accurately identify non-DL patients. Aside from the imaging airway test, other bedside airway tests have also been assessed in other systematic reviews [5,41,42]. Both MMT and ULBT showed a relatively high specificity of 84% [41] and 92% [5], respectively, but both tests showed relatively poor results for sensitivity (MMT 55% [41] and a ULBT of 67% [5]. Another meta-analysis on ULBT shared similar results [42]. Other bedside airway tests examined by Roth et al. [5], including a Wilson risk score, TMD, sternomental distance, and mouth opening, all displayed the similar pattern, i.e., a high specificity but a poor sensitivity. With similar, if not higher sensitivity and specificity, our results proved that TMHT is a rather comprehensive single predictor for DL, as most of the other predictors showed an unbalanced relation between sensitivity and specificity in other meta-analysis [5].

Last but not least, a prespecified threshold value plays an important role in reducing bias, and leads to a more impartial result [30]. Thus, in our study, a subgroup analysis containing 10 studies [14,15,16,20,23,24,34,35,38,39] with prespecified TMHT thresholds was conducted and showed a great predictive value by hitting the highest level of all tests and studies, with a summary sensitivity of 82% (95% CI, 71–89%), a specificity of 94% (95% CI, 87–98%) (Appendix A), and an AUC of 92% (95% CI, 90–94%) (Appendix A). That is to say, after reducing the existing bias to a certain degree, the outstanding predictive values from subgroup analysis were those that best represented the actual reliability of TMHT in predicting DL, confirming its excellent predictive potential.

The meta-analysis conducted by Carvalho et al. [21] in 2021 reported similar but somewhat limited results. In their analysis, summary sensitivity and specificity for studies with a common threshold were 82.6% (95% CI, 74–88.8%) and 93.5% (95% CI, 79–98.2%), respectively [21]. Obviously, there are numeral differences in both sensitivity and specificity between their analyses and ours. However, what needs to be kept in mind is the significant enlargement for the number of studies and total sample size/range included in our analysis. A total of 23 studies with 5896 patients were included, almost doubling the sample size, compared to 8 studies with 2844 patients as reported by Carvalho et al. [21]; especially, three of the most recent studies [22,25,26], which are all included, are aimed at the predictive performance of TMHT for DL in specific populations with risk factors of difficult airways, such as obesity, an age over 65 years, and others. Knowing that these factors are directly associated with the prevalence of DL and were considered as exclusion criteria in Carvalho et al.’s [21] work, the current study faces an extra challenge in the process of analysis and gains an extra validity in the results. Eight studies [20,33,34,35,36,37,38,39] conducted on the Chinese and Nepalese populations were included in our analysis, contributing to the ethnic diversity of patients. In total, the current study included 2355 Mongoloid subjects [18,33,34,35,36,37,38,39], 3225 Indian Mediterranean type Caucasian subjects [11,12,13,14,15,17,19,20,22,23,24,25,26,27], and 316 Baltic Sea type Caucasian subjects [16]. Baltic Sea type Caucasian subjects could be under-represented. No African type subjects were included in this meta-analysis. None of all 23 included studies in our analysis were eliminated during meta-analysis whether or not sharing the same threshold. These characteristics allowed a more comprehensive and representative population, bringing down the concern for bias and bolstering the credibility. Moreover, if a common 5 cm threshold was set, our data showed a summary sensitivity of 76% (95% CI, 66–83%) and a specificity of 91% (95% CI, 82–95%), almost at the same level with Carvalho et al.’s results [21]. Publication bias, not occurring in the current analysis, was suggested to be present in Carvalho et al.’s analysis [21], which also brought positive impact to their summary estimates. Thus, the results of the current study concurred with those of Carvalho et al.’ analysis [21] but take a step forward, i.e., providing a more valid proof for the ability of TMHT in predicting DL.

## 5. Limitations and Implications

In spite of the already impressive potential of TMHT, the summary estimates of TMHT for all 23 studies, sensitivity 74% and specificity 88%, in fact failed to reach expectation, possibly due to the impact of un-standardized application of external laryngeal manipulation across studies. External laryngeal manipulation, known as BURP, referring to external, backward, upward, and rightward pressure, can be applied when the designated airway assessor estimates the laryngeal view of the patient for the purpose of predicting difficult airway [18,22]. Helping the practitioner to get a better view, it is worthwhile to combine BURP with the CL grading system to better determine DL. However, the presence of BURP would modify the final CL grading, affecting the final determination of DL. Moreover, the lack of proper principles and consistent indications for BURP application in the included studies would have caused confusion in the screening process, as some studies applied BURP on all patients [22], while other studies applied it only on the second attempt [11,12] or only on poor CL grades of [16,18,20,36]. The consistency of the reference standard test CL grading was therefore perturbed, bringing significant heterogeneity. A fact worth mentioning is that the largest study included in this analysis also presented the lowest incidence of DL and the worst predictive performance of TMHT for DL among all included studies, with non-BURP evaluation showing a slightly better accuracy (68.1% versus 53.4%) [18]. This brought up the idea that the BURP manipulation might result in unintended stringent CL grading and conflict with TMHT, thus the predictive value of THMT in our analysis, on the whole, seemed unsatisfying when at least 9 [11,12,16,18,19,20,22,26,36] out of 23 studies mentioned the presence of BURP. That is to say, TMHT anticipates an even better performance in DL prediction whenever a consistent BURP policy is announced.

The ideal evidence for the current study, in fact, would be the studies with both low risk-of-bias and 5 cm threshold. Unfortunately, however, only two of the studies [13,17] included in this meta-analysis matched these conditions. Thus, there is not enough data for conducting such a sub-group meta-analysis. As TMHT is a novel single parameter and relevant study design still awaits improvement, we believe that more and more precise results about the predictive value of THMT for DL would be obtained in future studies and clinical practice. On the premise of the outcomes of current studies, with growing attention and more well-designed future clinical trials, TMHT may become a widely accepted indicator for prediction of DL among anesthesiologists.

## 6. Conclusions

Our analysis demonstrates that the predictive value of THMT for DL, on the whole, is more reliable than other imaging and bedside airway tests available in current practice. However, the significant heterogeneity and the uncertain influence brought by un-standardized BURP application indicate that further studies with a good design and a large sample size are still needed to determine the actual predictive value of TMHT for DL.

## Figures and Tables

**Figure 1 jcm-11-04906-f001:**
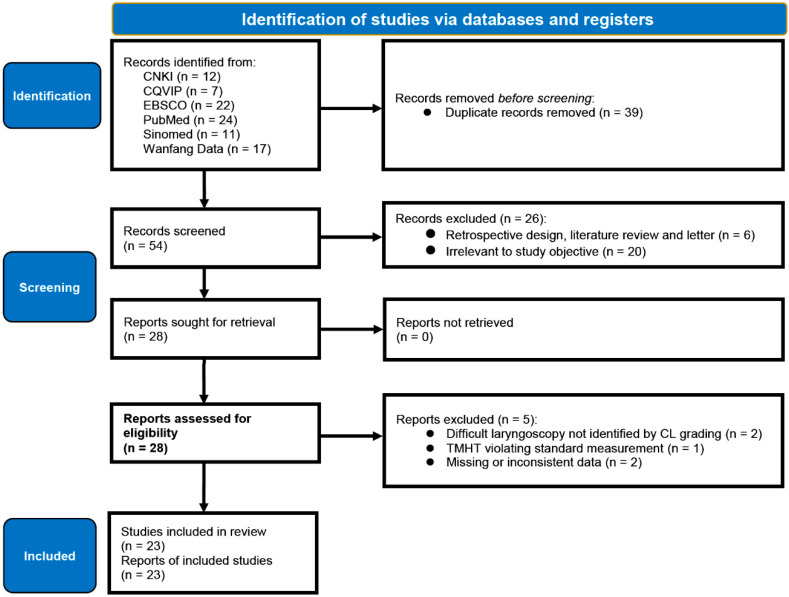
The PRISMA flow diagram of systematic review for included and excluded studies.

**Figure 2 jcm-11-04906-f002:**
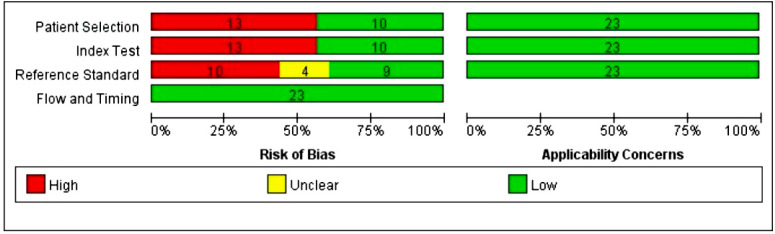
Risk of bias and applicability concerns review with authors’ judgments about each domain presented as percentages across included studies.

**Figure 3 jcm-11-04906-f003:**
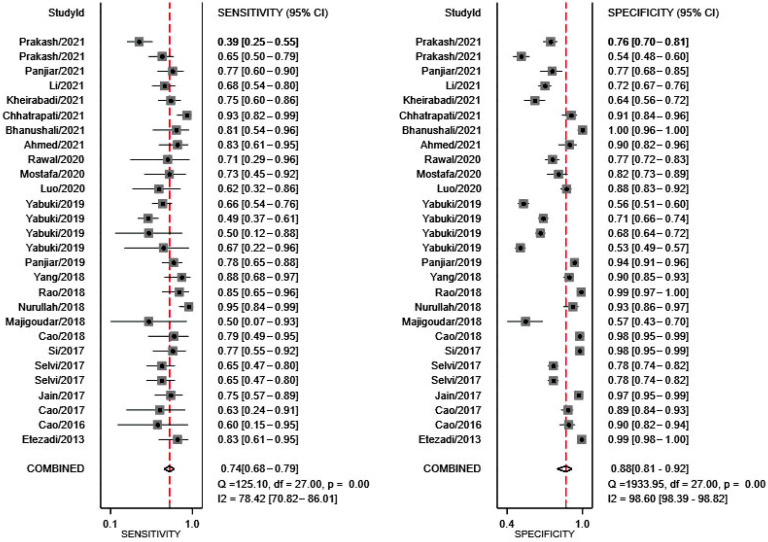
Forest plots of the analysis about the prediction value of TMHT for DL in terms of sensitivity and specificity with the data of all 23 studies. Square symbols represent the sensitivity or specificity of each study according to the Study ID shown on the y-axis, while the short lines cutting through represent the relative 95% CI. The diamond symbols refer to the combined sensitivity or specificity, which was automatically calculated and displayed by Stata software. A “COMBINED” label coordinating to the diamond symbol is shown on the y-axis underneath all Study ID.

**Figure 4 jcm-11-04906-f004:**
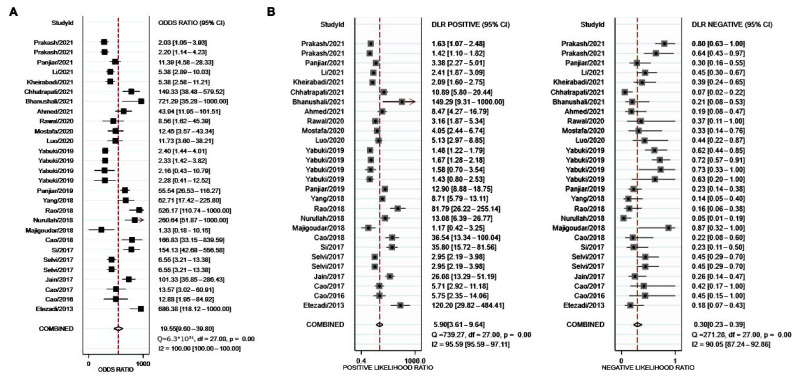
Forest plots of the analysis about the prediction value of TMHT for DL in terms of diagnostic odd ratio (DOR, (**A**)), positive likelihood ratio and negative likelihood ratio (LR+, LR−, (**B**)) with the data of all 23 studies. Square symbols represent the sensitivity or specificity of each study according to the Study ID shown on the y-axis, while the short lines cutting through represent the relative 95% CI. The diamond symbols refer to the combined sensitivity or specificity, which was automatically calculated and displayed by Stata software. A “COMBINED” label coordinating to the diamond symbol is shown on the y-axis underneath all Study ID.

**Figure 5 jcm-11-04906-f005:**
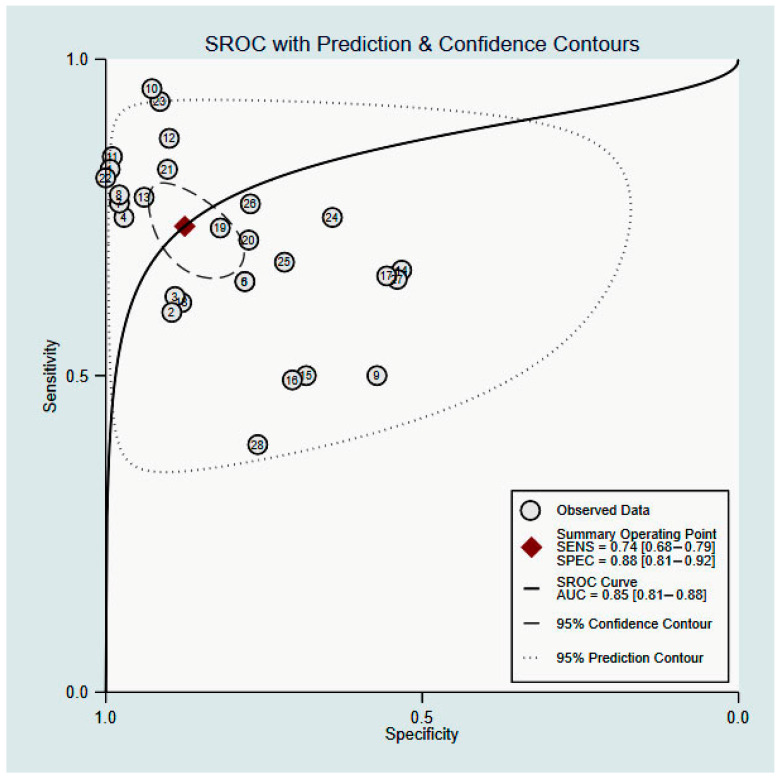
SROC of sensitivity and specificity of TMHT for prediction of DL with the data of all 23 included studies.

**Table 1 jcm-11-04906-t001:** Characteristics of included studies.

Authors	Years	Countries	Mean Age; Years	Male; %	Female; %	Mean Height; cm	Mean Weight; kg	Mean BMI	Total Sample Size	DL; n (%)	Thresholds; cm
Etezadi	2013	Iran	44.5	47.5	52.5	166.1	72.0	25.8	314	23 (7.3)	5
Cao	2016	China	43.0	56.7	43.3	NA	NA	24.2	120	5 (4.2)	5
Cao	2017	China	42.0	58.0	42.0	NA	NA	25.2	200	8 (4)	5
Jain	2017	India	56.7	NA	NA	162.6	65.3	24.7	345	32 (9.3)	5
Selvi	2017	Turkey	48.5	51.0	49.0	NA	77.7	NA	451	37 (8.2)	5
											4.35
Si	2017	China	51.4	NA	NA	165.0	NA	25.8	300	22 (7.3)	4.9
Cao	2018	China	44.6	56.0	44.0	NA	61.3	NA	200	24 (12)	4.9
Majigoudar	2018	India	39.8	53.3	46.7	NA	NA	21.3	60	4 (6.7)	5
Nurullah	2018	Bangladesh	45.4	50.4	49.6	NA	NA	NA	139	43 (31)	5
Rao	2018	Australia	43.4	47.2	52.8	162.6	62.0	23.4	316	26 (8.2)	5
Yang	2018	China	47.0	43.3	56.7	161.0	NA	23.0	263	24 (10)	3.92
Panjiar	2019	India	37.2	43.6	56.4	158.4	61.1	24.5	550	55 (10)	5
Yabuki	2019	Japan	50.2	18.0	82.0	159.6	58.6	22.9	609	6 (1)	5 with BURP
											5.4 with BURP
										73 (12)	5 without BURP
											5.4 without BURP
Luo	2020	China	49.9	38.4	61.6	160.6	62.4	NA	263	13 (4.9)	3.9
Mostafa	2020	Egypt	68.0	57.0	43.0	NA	NA	27.1	120	15 (12)	5.7
Rawal	2020	Nepal	35.8	44.3	55.7	158.0	60.9	24.1	246	7 (2.8)	5
Ahmed	2021	Egypt	38.3	78.1	21.9	NA	NA	43.7	105	23 (21.9)	4.7
Bhanushali	2021	India	51.7	40.4	59.6	162.4	NA	NA	109	16 (14.7)	5
Chhatrapati	2021	India	36.8	53.3	46.7	NA	55.2	NA	150	50 (30)	5
Kheirabadi	2021	Iran	41.3	32.1	67.9	NA	NA	35.7	196	48 (24.5)	4.8
Li	2021	China	NA	52.0	48.0	NA	NA	NA	400	53 (13.25)	4.805
Panjiar	2021	India	69.4	48.4	58.6	154.1	54.2	23.1	140	35 (25)	5.5
Prakash	2021	India	40.9	60.7	39.3	162.4	60.3	22.9	300	46 (15.3)	5
											4.4

BMI: body mass index (kg/M^2^); DL: difficult laryngoscopy.

## Data Availability

The study does not report any data.

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
