# Peer review of "Use of the Thyromental Height Test for Prediction of Difficult Laryngoscopy: A Systematic Review and Meta-Analysis"

_jcm, 2022, doi:10.3390/jcm11164906_

Round 1
Reviewer 1 Report
This systematic review and meta-analysis aimed to offer insight into the predictive usefulness of thyromental height test (a new clinical test for predicting difficult laryngoscopy, that measures the distance, with the patient in supine position and mouth closed, between the anterior border of the thyroid cartilage (on the thyroid notch just between the two thyroid laminae) and the anterior border of the mentum) in predicting difficult laryngoscopy.
Excellent systematic review with meta-analysis, as utility, structure, methodological construction, statistical analysis, and clarity!
Important issues
Please make a subgroup meta-analysis only on the low risk-of-bias studies with the 5 cm threshold (those without selection and index test biases), and present its results. This would be of a great utility for the audience of this paper. In fact it will be the best evidence on the topic.
Minor issues
Please create a Limitation section in the Discussion chapter, where the issues with the quality of the articles, heterogeneity, and other limitations should be mentioned
Maybe a section Implications for clinical practice and future research would be useful, in the Discussion chapter
Results
page 11, line 12: ”The identical analytical procedure was conducted again after removing two studies of heterogeneity”, Please rephrase.
Author Response
- The point-by-point responds to Reviewer 1’ suggestions.
(1) Question 1: Important issues: Please make a subgroup meta-analysis only on the low risk-of-bias studies with the 5 cm threshold (those without selection and index test biases), and present its results. This would be of a great utility for the audience of this paper. In fact it will be the best evidence on the topic.
Response: Thank you for your question. Indeed, it was a great and inspiring idea to make a subgroup meta-analysis only on the low risk-of-bias studies with the 5 cm threshold (those without selection and index test biases). In available literature, however, there are only two studies that meet the requirements of subgroup analysis. Because of a significant heterogeneity among the included studies, moreover, we have actually performed a sensitivity analysis by excluding 2 outliers’ studies. Thus, it may be not of great value to make a sub-group analysis for only two studies. In the revised paper, this issue has been included in the Limitations and implications section for specific attention.
(2) Question 2: Minor issues: Please create a Limitation section in the Discussion chapter, where the issues with the quality of the articles, heterogeneity, and other limitations should be mentioned. Maybe a section Implications for clinical practice and future research would be useful, in the Discussion chapter
Response: Thank you for your suggestion. In this revision, the contents about limitations and implications of our analysis have been added. Furthermore, the issues with heterogeneity and other limitations have also been mentioned and pointed out.
(3) Question 3: Results: page 11, line 12: ”The identical analytical procedure was conducted again after removing two studies of heterogeneity”, Please rephrase.
Response: Thank you for your suggestion. In this revision, the contents of line 12 in page 11 have been rephrased.
Reviewer 2 Report
The study is a meta-analysis aiming to determine the actual performance of thyromental height test (TMHT) as a single predictor of difficult laryngoscopy (DL). A few studies have explored the performance of TMHT to predict DL. Although a recent meta-analysis exploring exactly the same subject was published by Carvalho et al. in 2021 (Carvalho CC et al. Braz J Anesthesiol. 2021: 0104-0014(21)00271-2), the authors decided to perform this new meta-analysis. The authors have included more studies (23) than Carvalho (8) because of recent publications and because of the inclusion of studies published in English, Portuguese and Chinese. The results of both meta-analyses were close. Because of heterogeneity, the authors have performed a sub-group analysis by excluding 2 outliers’ studies. This sub-group analysis reinforces the results. However significant heterogeneity was still present. The authors conclusions were “Our analysis demonstrates that the predictive value of THMT for DL, on the whole, is more reliable than other imaging and bedside airway tests, encouraging it to be performed as a routine airway assessment. However, the significant heterogeneity and the uncertain influence brought by un-standardized BURP application still need careful examination. Further studies on relations among BURP, CL grading, and TMHT would help us better understand the actual predictive potential of TMHT.”
The text is well written, easy to read. The methodology is good, the meta-analysis and the statistical analyses are correctly carried. However, I have some questions and comments:
1) all the information sources should be described. In the present meta-analysis did the authors search the grey literature? Did they contact directly study authors? Did they search unpublished studies?
2) the authors said that publications in Portuguese were included. Were the authors able to read in Portuguese? If not, how they did deal with Portuguese?
3) the criteria “difficult laryngoscopy” was defined by the grades 3 or 4 of Cormack & Lehane grading system. That means that the glottis was difficult to see, but not that the orotracheal intubation was difficult. In fact, only the difficulty to intubate is life-threating. This should be discussed as a limit of the study.
4) external laryngeal manipulations (BURP) improve the glottis exposition (Tamura M et al. Anesthesiology 2004 Mar;100(3):598-601. / Takahata O et al. Anesth Analg 1997 Feb;84(2):419-21.). Thus, the Cormack & Lehane grade is modified when BURP is applied. In our opinion, not to be aware of the BURP use for each included study is a major bias of the meta-analysis.
5) the calculated heterogeneity (I2) is very high. The sub-group analysis heterogeneity is still high. This should encourage the authors to be very careful with the conclusions drawn from this meta-analysis. In our opinion, the authors are not enough careful in the discussion and with their conclusions.
6) it has been previously shown that scores including multiples parameters are more performant to predict difficult intubation than single parameter. Other parameters, as Mallampati grade, mouth opening or cervical rachis mobility for instance, are easy to collect. Why have the authors chosen to evaluate the performance of a single parameter rather a score including multiples parameters? Why have the authors chosen to compare, in the discussion, single parameters together? This should be discussed.
7) The studies included in this meta-analysis are from China (7), Iran (2), Nepal (1), India (7), Bangladesh (1), Japan (1), Egypt (2), Turkey (1), and Australia (1). Caucasian subjects are probably under-represented in this meta-analysis. This should be discussed.
Author Response
- The point-by-point responds to Reviewer 2’ and 3 comments.
Comments: The study is a meta-analysis aiming to determine the actual performance of thyromental height test (TMHT) as a single predictor of difficult laryngoscopy (DL). A few studies have explored the performance of TMHT to predict DL. Although a recent meta-analysis exploring exactly the same subject was published by Carvalho et al. in 2021 (Carvalho CC et al. Braz J Anesthesiol. 2021: 0104-0014(21)00271-2), the authors decided to perform this new meta-analysis. The authors have included more studies (23) than Carvalho (8) because of recent publications and because of the inclusion of studies published in English, Portuguese and Chinese. The results of both meta-analyses were close. Because of heterogeneity, the authors have performed a sub-group analysis by excluding 2 outliers’ studies. This sub-group analysis reinforces the results. However significant heterogeneity was still present. The authors conclusions were “Our analysis demonstrates that the predictive value of THMT for DL, on the whole, is more reliable than other imaging and bedside airway tests, encouraging it to be performed as a routine airway assessment. However, the significant heterogeneity and the uncertain influence brought by un-standardized BURP application still need careful examination. Further studies on relations among BURP, CL grading, and TMHT would help us better understand the actual predictive potential of TMHT.”
Response: Thank you for your detailed comments on our work. We completely agree with you. The significant heterogeneity among included studied, a common problem of meta-analysis, is indeed an issues of concern in our study, though we had attempted to apply the current statistical methods to solve this problem as soon as possible. The results of our meta-analysis were same as previous work of Carvalho et al, but inclusion of more recent works is a power of our analysis. This at least suggests that TMHT is a potentially useful bed test for prediction of DL that deserves further study. In this revision, we have further clarified this problem and provided possible solutions in the limitations and implications of discussion section.
- The point-by-point responds to Reviewer 3’s questions and comments.
(1) Question 1: All the information sources should be described. In the present meta-analysis did the authors search the grey literature? Did they contact directly study authors? Did they search unpublished studies?
Response: Thank you for your question. In performing the present meta-analysis, we had searched the grey literature, but not result. One study ([37] Li M. The effectiveness of ultrasound measurement of upper airway anatomical parameters in predicting difficult airway. Electron J Master's Thesis. 2021. doi: 10.27162/d.cnki.gjlin.2021.005983) was a Chinese master’s thesis paper exclusively available online at CNKI (in Chinese), but we can obtain all data required for our analysis. All other studies included in the present meta-analysis are white literature and we can get the needed data of our analysis without the help of the authors as all the required information was clearly described in their studies.
(2) Question 2: The authors said that publications in Portuguese were included. Were the authors able to read in Portuguese? If not, how they did deal with Portuguese?
Response: Thank you for your question. We are able to read the literature in Portuguese and there is one Portuguese publication found during literature search. However, it was excluded during screening due to its different difficult laryngoscopy grading system and was not included in the final meta-analysis. We have revised section 3.1 to make sure that this exclusion was clear.
(3) Question 3: The criteria “difficult laryngoscopy” was defined by the grades 3 or 4 of Cormack & Lehane grading system. That means that the glottis was difficult to see, but not that the orotracheal intubation was difficult. In fact, only the difficulty to intubate is life-threating. This should be discussed as a limit of the study.
Response: Thank you for your comment. Although difficult laryngoscopy does not equal to difficult intubation, it is still an important warning for high risk of difficult or failed intubation. CL grading system, the TMHT and many other multi-parameters are all aiming to effectively predict difficult laryngoscopy, which then provide a signal for anesthesiologists to keep in mind while intubation. According to your suggestion, we have also added such clarification in the Introduction section.
(4) Question 4: External laryngeal manipulations (BURP) improve the glottis exposition (Tamura M et al. Anesthesiology 2004 Mar;100(3):598-601. / Takahata O et al. Anesth Analg 1997 Feb;84(2):419-21.). Thus, the Cormack & Lehane grade is modified when BURP is applied. In our opinion, not to be aware of the BURP use for each included study is a major bias of the meta-analysis.
Response: Thank you for your comments. We agree with you that external laryngeal manipulations (BURP) improve the glottic exposure and the unstandardization of BURP use is a major source of bias for the present meta-analysis. According to requirements of GCP for difficult airway management study, however, the researchers should report the classification of best glottic exposure during laryngoscopy, even if the BURP is used. Thus, we believe that the classifications of glottic exposure during laryngoscopy in all the studies included in this analysis to define DL should be best glottic exposure that the researchers can obtain. Of course, the influence of nonstandardized BURP use on the glottic exposure is not excluded, as both human factors and operation levels are very difficult to control in clinical trial.These issues have been described in the limitations of Discussion section.
(5) Question 5: The calculated heterogeneity (I2) is very high. The sub-group analysis heterogeneity is still high. This should encourage the authors to be very careful with the conclusions drawn from this meta-analysis. In our opinion, the authors are not enough careful in the discussion and with their conclusions.
Response: Thank you for your good question. Just like we have described in discussion, the high calculated heterogeneity (I2) indeed is a concern on the results of this analysis. Thus, we have carefully discussed this issues and rewritten the conclusions as following.
Our analysis demonstrates that the predictive value of THMT for DL, on the whole, is more reliable than other imaging and bedside airway tests available in current practice. However, the significant heterogeneity and the uncertain influence brought by un-standardized BURP application indicates that further studies with good designs and large sample size are still needed to determine the actual predictive value of TMHT for DL.
This will help to remind the readers that they should cautiously use the TMHT for prediction of DL.
(6) Question 6: It has been previously shown that scores including multiples parameters are more performant to predict difficult intubation than single parameter. Other parameters, as Mallampati grade, mouth opening or cervical rachis mobility for instance, are easy to collect. Why have the authors chosen to evaluate the performance of a single parameter rather a score including multiples parameters? Why have the authors chosen to compare, in the discussion, single parameters together? This should be discussed.
Response: Thank you for your questions. Indeed, multi-parameters have been used as the predictors for difficult laryngoscopy in the current clinical practice as there has not been an effective single parameter discovered. The multi-parameters happened to be the best choice so far, but multi-parameters could inevitably bring more bias during each measurement and prediction. As mentioned in the discussion, we believe that TMHT, as a newly proposed single parameter, is showing a better predictive performance than the multi-parameters available in current practice. Thus, it is worth evaluating whether the TMHT is truly a more simple and effective predictor for difficult laryngoscopy.
(7) Question 7: The studies included in this meta-analysis are from China (7), Iran (2), Nepal (1), India (7), Bangladesh (1), Japan (1), Egypt (2), Turkey (1), and Australia (1). Caucasian subjects are probably under-represented in this meta-analysis. This should be discussed.
Response: Thank you for your comments. In this revision, we have added possible influence of under-represented Caucasian subjects on the results of this meta-analysis in the discussion section.
Round 2
Reviewer 1 Report
Thank you for the improvements and for the answers!
No further comments.
Author Response
Thank you so much for your previous inspiring questions and comments in helping us to improve the manuscript. We truly appreciate all of your time and effort put on our paper.
Reviewer 2 Report
We would like to thank author for the improvement of the text. There is only a minor comment: no study is dealing with african-type subjects among the studies included in the present MA. We think this should be added in the discussion as follow : " In total, the current study included 2355 Mongoloid subjects [18,33-39], 3225 Indian Mediterranean type Caucasian subjects [11-15,17,19,20,22-27], and 316 Baltic Sea type Caucasian subjects [16]. Baltic Sea type Caucasian subjects could be under-represented. No African-type subjects were included in this meta-analysis."
Author Response
Response: Thank you so much for your comment. Today, we have conducted another round of literature search and no new study is found. Thus, our meta-analysis would have to end up with no African-type subjects. This issue has been added to the Discussion section as a limitation. This sentence is signed by yellow word.
According to your suggestions, we have checked and confirmed that all references are relevant to the contents of the manuscript.
